# Characterisation of the Offshore Precipitation Environment to Help Combat Leading Edge Erosion of Wind Turbine Blades

Robbie Herring[1], Kirsten Dyer[1], Paul Howkins[1], Carwyn Ward[2]

[1]Offshore Renewable Energy Catapult, Offshore House, Albert Street, Blyth, NE24 1LZ, UK

[2]Department of Aerospace Engineering, Queen's Building, University of Bristol, Bristol, BS8 1TR, UK

*Correspondence to*: Robbie Herring (robbie.herring@ore.catapult.org.uk)

**Abstract.** Greater blade lengths and higher tip speeds, coupled with a harsh environment, has caused blade leading edge erosion to develop into a significant problem for the offshore wind industry. Current protection systems do not last the lifetime of the turbine and require regular replacement. It is important to understand the characteristics of the offshore

environment to model and predict leading edge erosion. The offshore precipitation environment has been characterised using up to date measuring techniques. Heavy and violent rain was rare and is unlikely to be the sole driver of leading edge erosion. The dataset was compared to the most widely used droplet size distribution. It was found that this distribution did not fit the offshore data and that any lifetime predictions made using it are likely to be inaccurate. A general offshore droplet size distribution has been presented that can be used to improve lifetime predictions and reduce lost power production and

unexpected turbine downtime.

## 1 Introduction

The offshore wind industry's need of larger rotors and higher tip speeds has caused blade leading edge erosion to develop into a major problem for the industry. Leading edge erosion is caused by raindrops, hailstone, and other particles impacting the leading edge of the blade and removing material. This degrades the aerodynamic performance of the blade and requires

operators to perform expensive repairs. The issue has grown in prominence recently with reports that Ørsted had to make repairs to up to 2,000 offshore wind turbines after just a few years of operation (Finans, 2018).

The industry attempts to prevent the onset of leading edge erosion by applying protection systems, such as coating and tapes, to the blade leading edge. However, currently these do not last the lifetime of the turbine and require regular replacement.

Several analytical models that aim to estimate the expected lifetime of a protection system have been developed (Eisenberg

et al., 2018, Slot et al., 2015, Springer et al., 1974). Finite element models that can predict the stresses and strains in a protection system from an impinging water droplet have also been produced (Keegan et al., 2012, Doagou-Rad and Mishnaevsky, 2019). To model leading edge erosion, it is important to understand the characteristics of the impinging hydrometeors and, as rain is the most frequent hydrometeor, the droplet size distribution (DSD) of the impinging rain.

The aim of the industry is to develop a methodology that can predict the lifetime of a protection system on a wind turbine from rain erosion tests. The DNV-GL project COBRA aims to address this, and Eisenberg proposes using the Springer model. Due to the lack of an offshore dataset, the project uses the onshore Best distribution published in 1950 (Best, 1950). However, the manual measurement techniques used by Best are outdated and have been found to provide inaccurate results (Kathiravelu et al., 2016). The lack of an offshore dataset introduces uncertainty into lifetime predictions and, as a result,

inaccuracies may exist. In this work, state of the art measurement techniques have been used to characterise the offshore precipitation environment and provide the required offshore dataset. A general offshore DSD is presented.

## 2 The Best Distribution

The most widely used DSD is the Best distribution. Best takes the work of several authors and converts them into a common DSD defined as:

$$1 - F = exp\left[-\left(\frac{x}{a}\right)^n\right], \tag{1}$$

where $F$ is fraction of liquid water in the air comprised by drops with diameter less than $x$, $I$ is the rate of precipitation and

$$a = AI^p, \tag{2}$$

where $A = 1.30, p = 0.232, n = 2.25$. Best concluded that the constant $n$ is independent of the precipitation intensity. This is commonly presented in literature as:

$$F(x) = 1 - exp\left[-\left(\frac{x}{1.3\ I^{0.232}}\right)^{2.25}\right], \tag{3}$$

Data was predominantly collected by two manual methods; the 'Stain' method and the 'Flour Pellet' method. In the Stain method, a sheet of absorbent paper is exposed to the rain for a short time. The stains made by the droplets are rendered permanent by previously treating the paper with a suitable powder dye. Then, the stains are counted, measured and interpreted in terms of drop sizes. A calibration curve specific to the filter paper is used to relate the stain diameter to the

droplet diameter. The spread factor relationship is dependent upon the physical properties of the fluid, drying conditions and the impact velocity of the droplet (Sommerville and Matta, 1990). In the Flour Pellet method, rain is allowed to fall into pans of silted flour. The resulting dough pellets are baked and subsequently sized by passing them through graded sieves.

In both measurement techniques, sampling can only occur in short intervals. Best performs measurements using the Stain method for a maximum of two minutes. During prolonged periods of sampling, the droplet stains and pellets can overlap,

making it difficult to accurately measure and count individual drops. Furthermore, the techniques also have a low resolution. Best registers droplet sizes in 0.5 mm intervals. Given that the distribution predicts that for a rain rate of 1 mm/hr, most droplets are between 0 and 2 mm, it is clear that a higher resolution is required for effective analysis.

## 3 Offshore Measurement Technique

Two Campbell Scientific PWS100 disdrometers have been installed onto Offshore Renewable Energy Catapult's offshore anemometry hub, which is located three nautical miles from the coast of Blyth, Northumberland. Fig. 1 shows the position of the two disdrometers, with the first mounted on the existing platform 25 m metres above sea level (disdrometer A) and the second mounted 55 m above sea level (disdrometer B). Each disdrometer consists of two photodiode sensing heads, one near-IR diode laser head and one CS215 temperature and humidity sensor. The sensor heads are positioned 20° off-axis to the system unit axis, introducing a time-lag between the two sensors that enables the fall velocity and size of particles to be calculated.

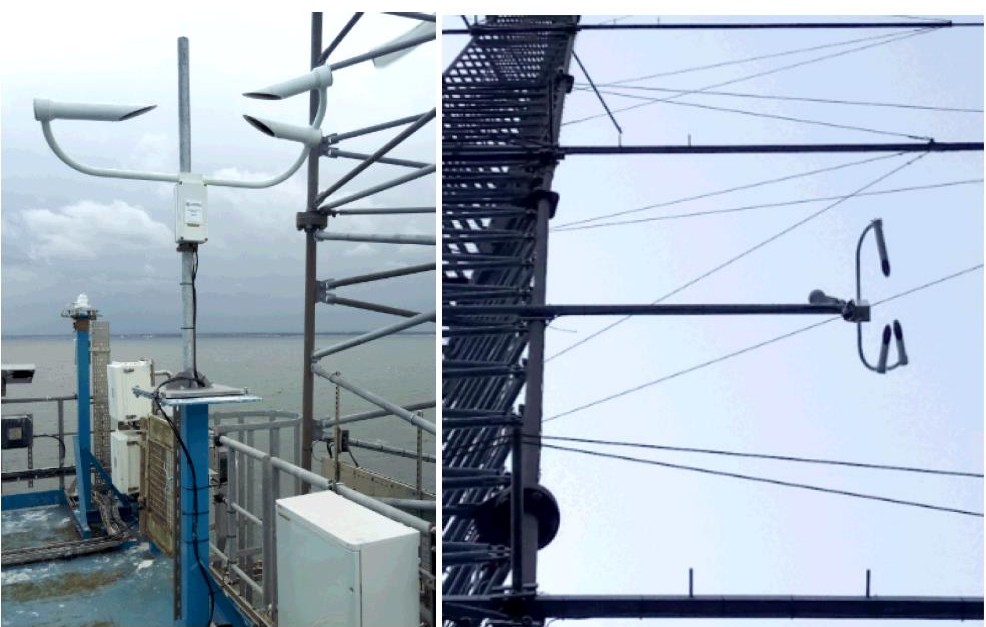

**Figure 1: The optical disdrometers mounted to the platform (left) and at 55 m above sea level (right).**

Optical disdrometers are non-intrusive and do not influence drop behaviour during measurement. They have also been shown to successfully resolve droplet break-up and splatter problems experienced by other measurement techniques (Kathiravelu et al., 2016). Agnew (Agnew, 2013) explored the performance of the PWS100 at a site in Southern England, finding that the device slightly underestimates the number of droplets with a diameter below 0.8 mm. However, the measurement of larger, more damaging droplets was found to be accurate. Montero-Martinez (Montero-Martínez et al., 2016) compared the performance of the disdrometer during natural rain events in Mexico City to results from a beam occlusion disdrometer and a reference tipping bucket. The PWS100 recorded greater amounts of precipitation than the reference, but the study was unable to back this up statistically and no significant differences in precipitation estimation was found between the disdrometers. Montero-Martinez concluded that the two devices performed similarly and that the

PWS100 provides reliable precipitation measurements. Johannsen (Johannsen et al., 2020) studied the PWS100 against a Thies Clima Laser Precipitation Monitor and a Parsivel OTT at a site in Austria. In contrast to Montero-Martinez, the PWS100 recorded less than the reference rain gauge in all but two events. The PWS100 recorded 3% less total precipitation that the rain gauge across the measurement period, outperforming the Thies and the Parsivel which recorded 20 and 30% less, respectively, and the PWS100 was consistently closest to the rain gauge reading throughout the period. Similar drop sizes were recorded between the PWS100 and the Parsivel, with Johannsen noting that the PWS100 tended to record slightly faster and larger drops. The studies show that there are uncertainties in the accuracy of all disdrometers, with the PWS100 used in this study performing comparatively or better than the other examined disdrometers.

DSD data from 1$^{st}$ September 2018 up to and including the 31$^{st}$ August 2019 is presented to provide a 12 month period for analysis. This allows analysis to also be completed seasonally. Hydrometeor diameters have been recorded with a resolution of 0.1 mm. Data is available with a time interval of 1 minute.

## 4 The Offshore Dataset

### 4.1 Quality Control

Raw data was received from the disdrometers and, therefore, detailed quality control was completed before subsequent analysis in line with recommendations from (Hasager et al., 2020), Chen (Chen et al., 2016) and Vejen (Vejen et al., 2018), Duplicate records were assessed by comparing time stamps, with any identical timestamps eliminated from the dataset. The meteorological parameters were also evaluated to remove entire duplicate records. It may be possible for a few parameters to be the same, however an entire row of identical parameters is extremely unlikely and consequently duplication has almost certainly occurred. A gross value check was completed to remove unrealistic and impossible values. Certain parameters are constrained within limits, such as relative humidity, which is given as a percentage, whereas other parameters, such as droplet size, can be evaluated against sensible threshold values. Furthermore, precipitation events where the disdrometer recorded a rain rate of 0 mm/hr, but hydrometeors were recorded were removed, as were events within the bounds of disdrometer error, such as those with a duration of 1 minute or where less than 10 total hydrometeors were recorded. Particle type classification is determined by the C215 sensor on the disdrometer, which distinguishes particles based on an algorithm using the temperature, wet bulb temperature and relative humidity. The outputs from the sensor were evaluated against an air temperature threshold, commonly used to distinguish between snow and rain events (Jennings et al., 2018), with any errors being manually inspected.

The consistency between disdrometers was also explored. No sensible results were recorded by disdrometer A from 23$^{rd}$ November 2018 until its repair at the start of May 2019, whilst disdrometer B remained operating throughout the year with short, infrequent gaps in data gathering. Of the available recordings, the two disdrometers agreed on the occurrence of precipitation 97.40% of the time, with this increasing to 99.74% when evaluating precipitation intensities above 0.5 mm/hr. Between the two disdrometers, 0.9% of the data recorded a difference in precipitation intensity greater than 1 mm/hr, with

these almost exclusively occurring in the higher precipitation intensities. A manual inspection of the greatest differences found that where large values were recorded in one disdrometer, the other recorded a comparable value in the surrounding minutes. This indicates that the large differences are correct and may suggest a small time discrepancy between the disdrometers, only noticed in the short, high intensity events.

The comparable data gathered by disdrometer A and B enabled some gaps in disdrometer B's dataset to be filled with the respective data from disdrometer A, where available. In total, 34.25 hours were gap filled, of which 229 minutes experienced precipitation and 111 minutes experienced a precipitation greater than 0.5 mm/hr.

Table 1 presents the percentage of available quality controlled data for each month and the percentage of the available data in which precipitation was recorded. An estimation of the actual percentage of precipitation can be obtained by assuming that the same proportion of precipitation occurred across the unavailable data. A total of 82.89% of the data was available during the entire measurement period. Precipitation was recorded in 8.71% of the available data giving a yearly precipitation estimate of 10.50%. Winter had the highest estimation of total time with precipitation with 12.07%, whilst spring saw the lowest with an estimation of 8.65%. Including the missing data provides an annual accumulation of 500 mm, which is lower than the 650 mm average annual precipitation reported in Northumberland (WeatherSpark, 2020), indicating that the measurement year was a relatively dry year for the area.

**Table 1: Percentage of available data for each month.**

| Month | Percentage of available values (%) | Percentage of time with precipitation (%) | Estimation of total time with precipitation (%) |
|---|---|---|---|
| September 2018 | 88.84 | 5.81 | 6.54 |
| October 2018 | 98.55 | 8.57 | 8.70 |
| November 2018 | 96.29 | 10.30 | 10.70 |
| December 2018 | 90.11 | 9.73 | 10.80 |
| January 2019 | 81.42 | 10.69 | 13.13 |
| February 2019 | 68.43 | 7.35 | 10.74 |
| March 2019 | 75.94 | 7.82 | 10.30 |
| April 2019 | 91.24 | 4.83 | 5.29 |
| May 2019 | 72.50 | 11.19 | 15.43 |
| June 2019 | 83.28 | 13.31 | 15.98 |
| July 2019 | 53.43 | 5.53 | 10.35 |
| August 2019 | 94.66 | 9.35 | 9.88 |
| Total | 82.89 | 8.71 | 10.50 |

## 4.1 Precipitation Intensity Frequency

The average precipitation intensity was recorded every minute. Fig. 2 presents its variation across the measurement period, and Fig. 3 presents the cumulative frequency of the recorded intensities. The median precipitation intensity for the measurement period was 0.311 mm/hr.

Precipitation is classified according to its intensity with the following categories defined by the Met Office (Met Office, 2007):

- Light – precipitation intensity less than 2.5 mm/hr,
- Moderate – precipitation intensity between 2.5 mm/hr and 10 mm/hr,
- Heavy – precipitation intensity between 10 mm/hr and 50 mm/hr,
- Violent – precipitation intensity greater than 50 mm/hr.

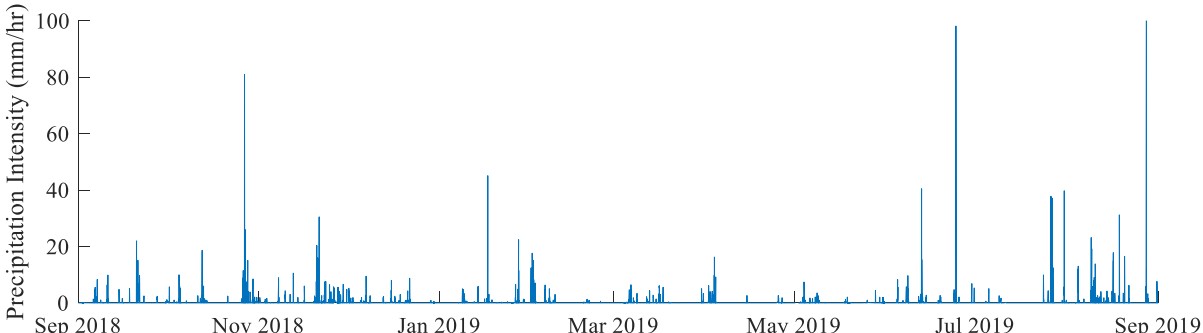

Figure 2: Precipitation intensity during the measurement period.

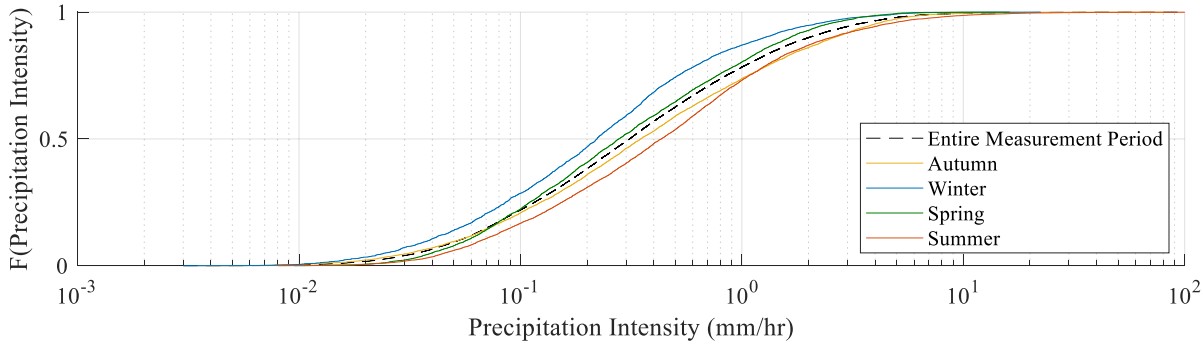

Figure 3: Cumulative distribution of precipitation for the respective seasons.

Table 2: Precipitation intensity distribution for seasons and intensity categories.

|  | Median precipitation intensity (mm/hr) | Percentage of precipitation category (%) | | | |
|---|---|---|---|---|---|
|  |  | Light | Moderate | Heavy | Violent |
| Autumn | 0.3492 | 89.42 | 10.09 | 0.46 | 0.03 |
| Winter | 0.2217 | 96.43 | 3.49 | 0.08 | 0 |
| Spring | 0.2778 | 98.56 | 1.44 | 0 | 0 |
| Summer | 0.4321 | 89.87 | 8.85 | 1.16 | 0.12 |
| Total | 0.3111 | 92.58 | 6.89 | 0.50 | 0.03 |

The seasonal breakdown of precipitation categories is shown in Table 2. Summer had the highest median precipitation intensity with the highest amount of recorded heavy and violent precipitation. In contrast, winter and spring saw minimal

heavy precipitation and no violent precipitation. Light precipitation dominated across the entire measurement period accounting for 92.58% of all precipitation. Furthermore, 78.31% of the recorded minutes had an intensity lower than 1 mm/hr. Moderate precipitation was recorded in 6.89% of all cases, whilst heavy and violent rain occurred in 0.50% and 0.03% cases, respectively. This corresponds to a total of 151 minutes of heavy precipitation and only 9 minutes of violent precipitation across the year. This gives a total of 193 minutes a year of heavy and violent rain once the unavailable data is factored in.

Therefore, a wind turbine in this location would experience less than 3.5 hours a year of precipitation with an intensity greater than 10 mm/hr. Without corresponding erosion data, it is not possible to conclude if erosion damage is predominantly caused by heavy and violent precipitation. However, given that erosion can occur within just a few years of installation and assuming that heavy and violent precipitation occurs with the same frequency as found in this dataset, a turbine would experience less than a day of high intensity rain before erosion occurs. When considering the Springer model, this suggests that erosion damage is not driven solely by heavy and violent precipitation disagreeing with current research theories (Bech et al., 2018).

**4.2 Hydrometeor Frequency**

Fig. 4 presents the number of recorded hydrometeors by type during the data collection period. The hydrometeor type is clearly dominated by rain droplets. 'Errors' and 'unknown' particles accounted for 17.93% of the hydrometeors recorded. These may be caused by insects, particles between states or equipment failures and have been ignored in the subsequent analysis, with any records where they were the modal hydrometeor removed. Drizzle and rain droplets make up a combined 98.45% of all hydrometeors recorded. The number of ice pellets, hail and graupel particles recorded was low, accounting for only 0.49% of hydrometeors recorded.

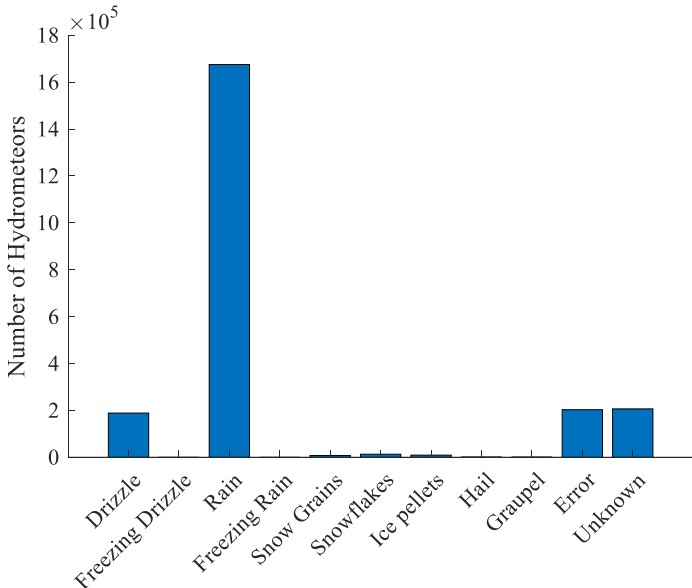

**Figure 4: Number and type of hydrometeors recorded during the total measurement period.**

As expected, ice and snow based hydrometeors occurred most frequently in winter. Ice pellets, hail and graupel accounted for 0.94% of the hydrometeors recorded in the season with snow grains and snowflakes accounting for 3.56%. In contrast, only 0.16% of hydrometeors recorded in summer were ice pellets or hail, with no graupel, snow grains or snowflakes. Spring and autumn respectively recorded 0.31% and 0.57% of ice pellets, hail and graupel.

### 4.3 Hydrometeor Velocity

The severity of a hydrometeor impact is governed by its kinetic energy. Whilst the blade speed provides most of the impact velocity, the hydrometeor fall velocity and mass are important. For each minute, the average diameter and velocity was plotted for the modal hydrometeor type. This is presented in Fig. 5.

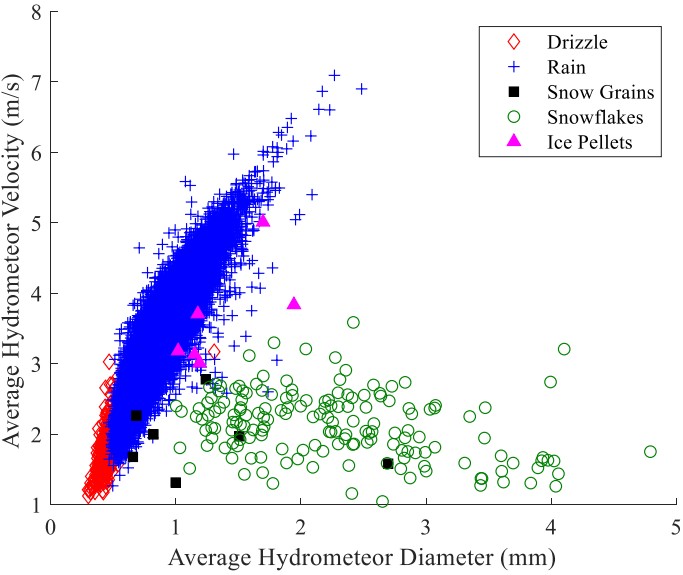

**Figure 5: Relationship between size and velocity for the modal hydrometeor at each minute.**

There is a clear distinction between water particles and snow particles, with snow particles occurring across a wider range of diameters and lower velocities than rain particles. For the few cases where ice pellets were the model hydrometeors, they all occurred to the right of the rain droplet scatter, indicating that they have a lower fall velocity that rain droplets. There were no cases where hail or graupel were the modal hydrometeor and they were found to be mixed in with rain particles. The presented velocities for water particles are in line with those predicted in models by Gossard (Gossard et al., 1992) and Brandes (Brandes et al., 2002). The data presented in the above figure is used in the subsequent analysis to estimate the number of droplets that impact the blade per second and inform lifetime prediction models.

## 5 Offshore Rain Distribution

To inform lifetime prediction models, a general equation for an offshore DSD is required. The Best DSD has been reproduced, both seasonally and non-seasonally, with updated constants for the offshore rain data presented. Only data where rain particles were the modal hydrometeor were examined.

### 5.1 Constant Derivation

For each recorded minute, the cumulative function, $F$, has been evaluated.
Rearranging Eq. (1) gives:

$$\ln \ln \left(\frac{1}{1-F}\right) = n \ln x - n \ln a ,\qquad(4)$$

Values of $n$ and $a$ for the average precipitation intensity over the minute can therefore be determined by plotting Eq. (4). Fig. 6 presents the evaluation of Eq. (4) across a range of precipitation intensities.

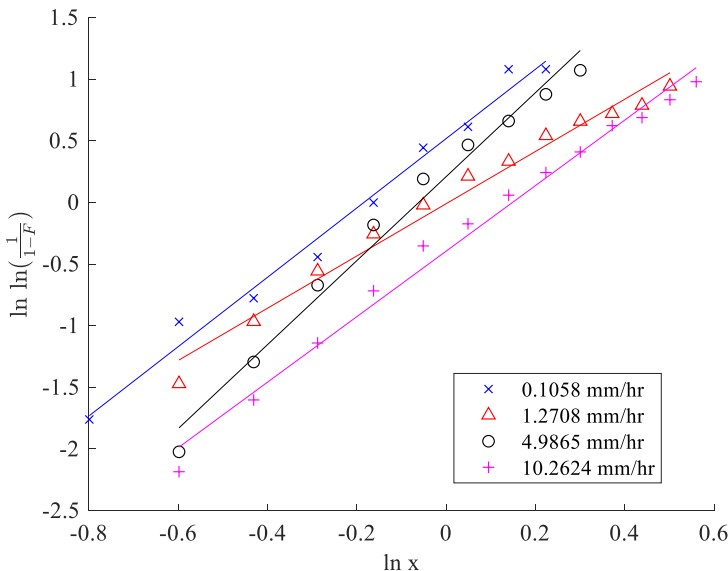

**Figure 6: Evaluation of Eq. (4) for precipitation intensities 0.1058, 1.2708, 4.9865 and 10.2624 mm/hr.**

Rearranging Eq. (2) gives:

$$\ln a = p \ln I + \ln A ,$$                                                                              (5)

By plotting Eq. (5), the constants $A$ and $p$ can be obtained. Fig. 7 evaluates Eq. (5) across the whole dataset.

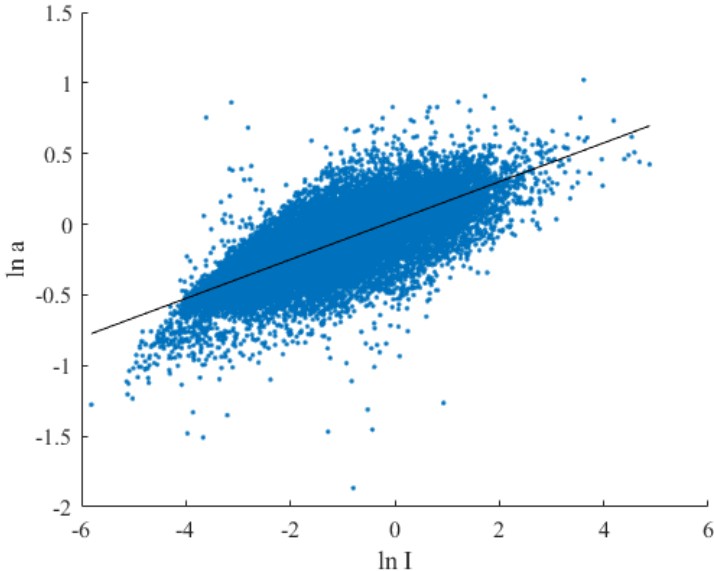

Figure 7: Evaluation of Eq. (5) to derive the constants $A$ and $p$.

The constants $A$ and $p$ are determined as 1.0260 and 0.1376, respectively.

Best concluded that the constant $n$ is independent of the precipitation intensity. However, for the data presented, $n$ has dependence on the rain rate. The following relationship applies:

$$n = NI^q , \tag{6}$$

This can be evaluated as:

$$\ln n = q \ln I + \ln N , \tag{7}$$

Fig. 8 presents the plot of Eq. (7) from which the constants $N$ and $q$ can be obtained.

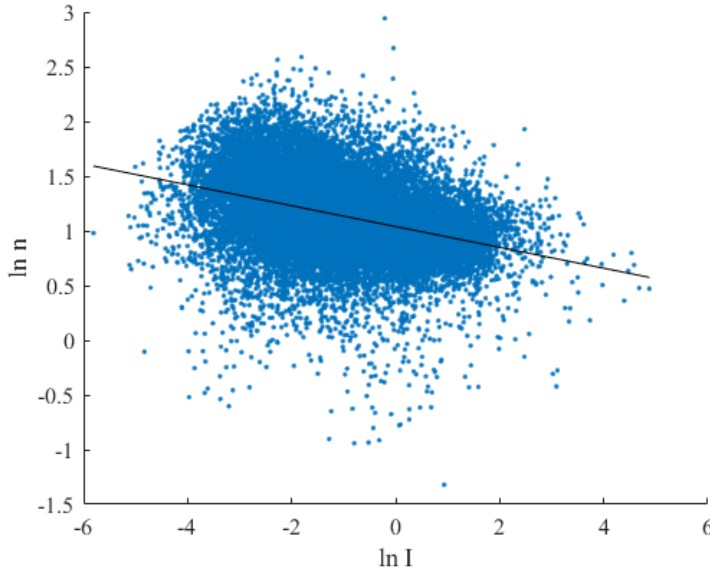

**Figure 8: Evaluation of Eq. (7) to derive the constants $N$ and $q$.**

The constants $N$ and $q$ are determined as 2.8264 and -0.0953, respectively. Fig. 8 shows substantial scatter in determining these constants. However, as $q$ is small there is only slight dependence of $n$ on the precipitation rate and whilst the scatter is likely to introduce some error, it does not have a significant effect on the resulting DSD. Table 3 summarises the constants for the non-seasonal distribution alongside the constants for seasonal DSDs. For detailed modelling and lifetime predictions it may be favourable to use season dependent DSDs.

**Table 3: Determined constants for the non-seasonal and seasonal offshore DSDs.**

| Season | Data used (%) | $A$ | $p$ | $N$ | $q$ |
|---|---|---|---|---|---|
| Non-seasonal | 100.00 | 1.0260 | 0.1376 | 2.8264 | -0.0953 |
| Autumn | 27.62 | 0.9723 | 0.1335 | 2.7762 | -0.0911 |
| Winter | 24.95 | 0.9831 | 0.1338 | 2.6581 | -0.1136 |
| Spring | 20.43 | 1.0393 | 0.1270 | 2.8282 | -0.1065 |
| Summer | 27.00 | 1.0937 | 0.1410 | 2.9657 | -0.0893 |

Reproducing Eq. (1) with the derived non-seasonal constants gives a general non-seasonal offshore DSD:

$$F(x) = 1 - exp\left[-\left(\frac{x}{1.03\ I^{0.138}}\right)^{\frac{2.83}{I^{0.0953}}}\right], \tag{8}$$

This is presented for various precipitation intensities in Fig. 9.

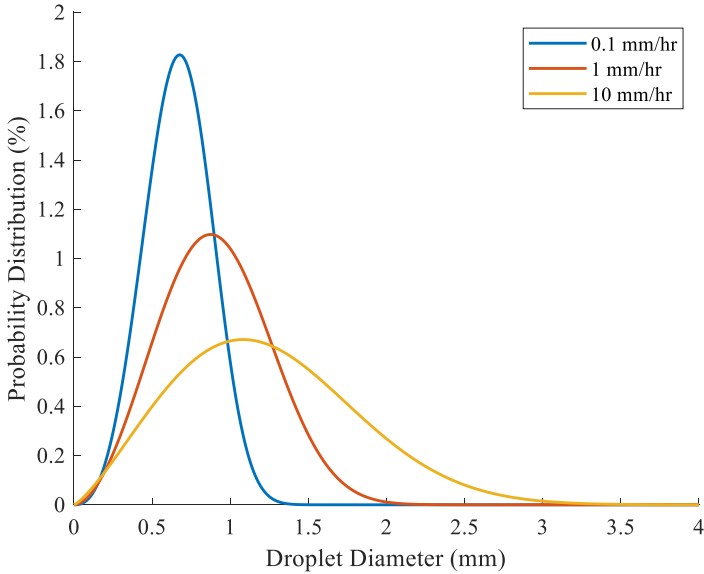


**Figure** 9**: The non-seasonal offshore DSD at different precipitation intensities.**

## 5.2 Sensitivity Analysis

The sensitivity of the constants to the data selected has been evaluated. The following cases have been examined:

- Low and high precipitation intensity have been individually and collectedly neglected. Precipitation intensities
below 0.1 mm/hr and above 10 mm/hr were neglected.

- Precipitation intensities that account for a small number of the recorded intensities have been individually and collectively neglected. These are the bottom 1% and the top 1%.

Minutes where the measured precipitation intensity is low generally record fewer droplets than those with higher precipitations. Conversely, a significant number of droplets are generally seen in heavy precipitation. Low and heavy
intensity rain may, therefore, have a high scatter that could influence the determined constants. Fig. 3 presented the cumulative distribution of the recorded precipitation intensities. The bottom and top 1% of precipitation intensities may also skew the data by providing a point significantly different to the trend. The impact of these conditions on the constants is shown in Table 4.

**Table 4: Sensitivity of constants to the selected cases.**

| Precipitation Intensities (mm/hr) | Data Used (%) | $A$ | $p$ | $N$ | $q$ |
|---|---|---|---|---|---|
| I | 100 | 1.0260 | 0.1376 | 2.8264 | -0.0953 |
| I > 0.1 | 77.68 | 1.0218 | 0.1249 | 2.8132 | -0.1067 |
| I < 10 | 96.85 | 1.0269 | 0.1382 | 2.8227 | -0.0961 |
| 0.1 < I < 10 | 6.89 | 1.0219 | 0.1252 | 2.8071 | -0.1090 |
| I > 0.0158 | 99 | 1.0245 | 0.1350 | 2.8223 | -0.0979 |

| | | | | | |
|---|---|---|---|---|---|
| I < 6.95 | 99 | 1.0280 | 0.1388 | 2.8192 | -0.0969 |
| 0.0158 < I < 6.95 | 98 | 1.0263 | 0.1360 | 2.8144 | -0.0997 |


In general, the constants are consistent across all the examined cases. The constant $p$ is the most sensitive to the data included. Neglecting low precipitation intensities reduces its value, whilst neglecting higher intensities increases its value. Removing precipitation intensities below 0.1 mm/hr has the greatest effect on the constants. However, ignoring these intensities loses 22.32% of the data available. It can be concluded that the proposed constants are acceptable.

**5.3 Comparison to Best DSD**

The general offshore DSD has been compared to the Best DSD at various precipitation intensities in Fig.10. The precipitation intensities 0.1, 1, 2.5, 5, 10, 20 mm/hr were selected to enable comparison of the two DSDs across a range of intensities. To account for variability in the recorded results, minutes which recorded an intensity within ±5% of the selected intensity were included. For each data group, the intensities were averaged and the offshore DSD and Best DSD for the

average intensity plotted against them.

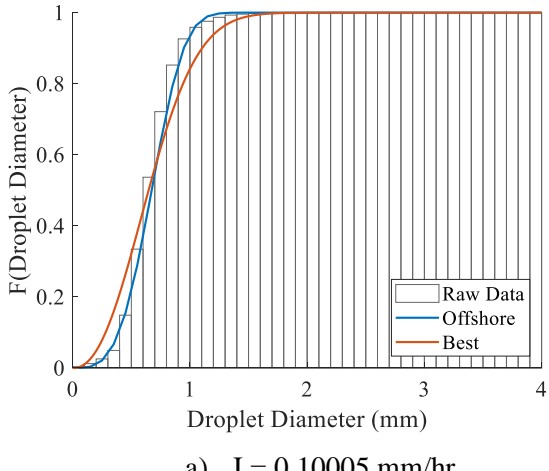 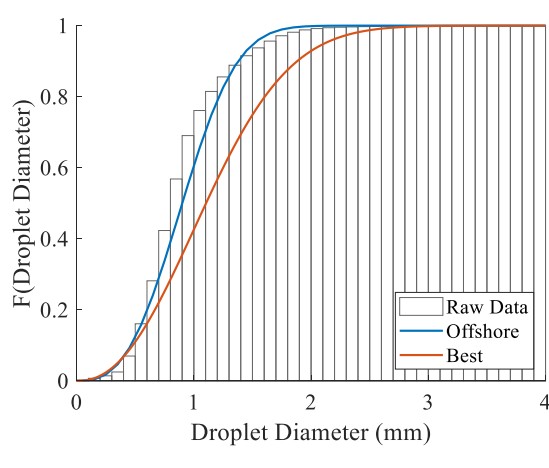

a)  I = 0.10005 mm/hr          b)  I = 0.99612 mm/hr

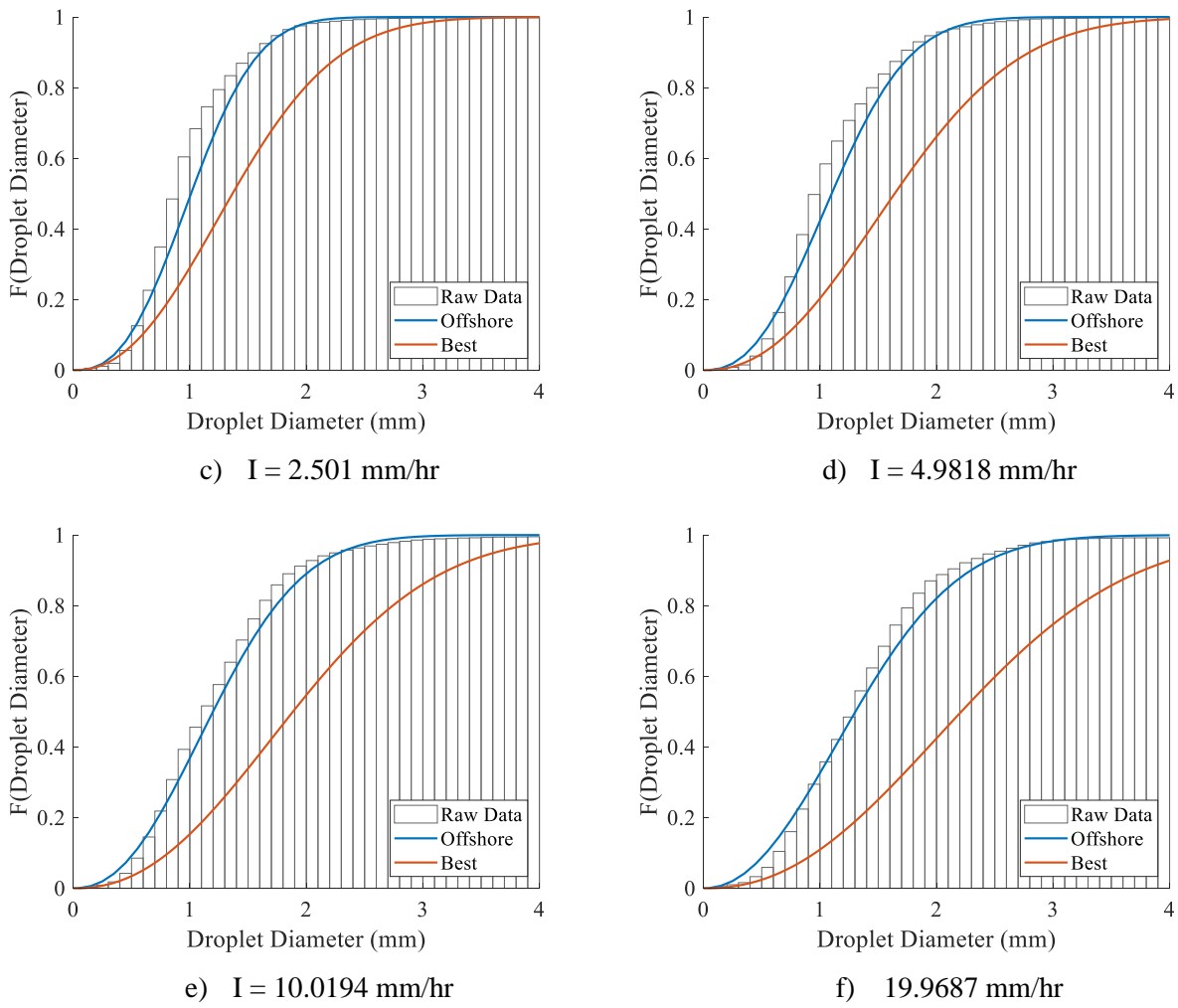

c)  I = 2.501 mm/hr

d)  I = 4.9818 mm/hr

e)  I = 10.0194 mm/hr

f)  19.9687 mm/hr

**Figure 10: Comparison between the offshore DSD and the Best DSD at precipitation intensities a) 0.10005, b) 0.99612, c) 2.501, d) 4.9818, e) 10.0194 and f) 19.9687 mm/hr.**

Fig 10. reveals that the Best DSD significantly overestimates the diameter of droplets. This is particularly true at the higher precipitation intensities. The goodness of fit of the offshore and Best DSD has been evaluated across the range of precipitation intensities in Fig. 11. The offshore DSD aligns well with the raw data and possesses a high coefficient of determination ($R^2$) across the precipitation intensity range. The slight reduction in $R^2$ at higher intensities can be attributed to the reduced amount of heavy and violent precipitation recorded. The coefficient of determination of the Best DSD reduces significantly as the precipitation intensity increases.


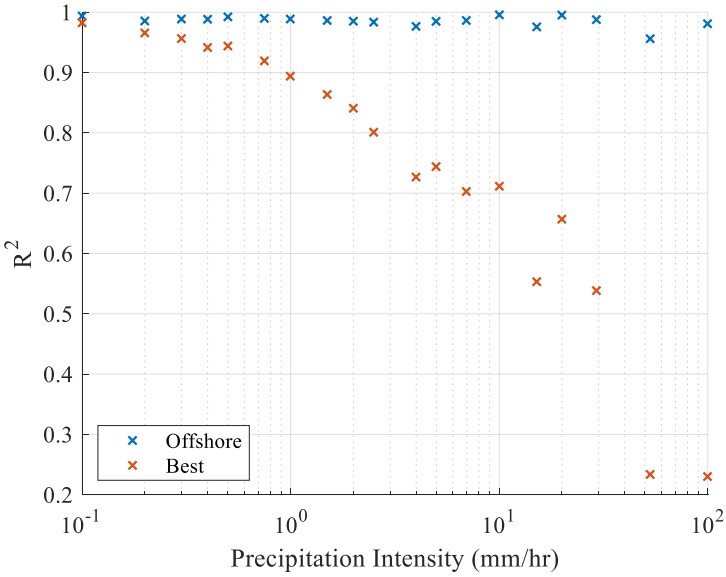

**Figure 11: Coefficient of determination of the offshore DSD and the Best DSD across a range of precipitation intensities.**

**5.4 Limitations**

The offshore DSD presented has two main limitations. Firstly, the presented measurement period may be a limiting factor. As the disdrometer continues to collect data, the DSD can be further refined. Secondly, data has only been collected at one point. Offshore DSDs may vary from location to location. To address this, a disdrometer has been positioned at ORE Catapult's Levenmouth offshore demonstration turbine for future comparison and validation.

**6 Impact of DSD on Leading Edge Erosion Lifetime Prediction**

The implications of the offshore DSD has been assessed using the Springer model, which is used by Eisenberg to predict a protection solution's in-situ lifetime from leading edge erosion. The model uses the median droplet diameter for a given rain rate to determine the number of impacts to failure, $N_{ic}$, and the number of impacts on the blade per m$^2$ per second, $\dot{N}$. The number of impacts to failure is found from:

$$N_{ic} = \frac{8.9}{d^2} \left(\frac{S_{ec}}{\overline{\sigma_o}}\right)^{5.7} , \tag{9}$$

where $S_{ec}$ is the effective strength of the protection system found from rain erosion test results, and $\overline{\sigma_o}$ is the pressure at the interface between the droplet and protection system, and is a function of the droplet diameter and the relative properties of the system to the substrate it is applied on. The number of impacts on the blade per m$^2$ per second is given as:

$$\dot{N} = q\, V_s\, \beta , \tag{10}$$

where q is the number of droplets in a cubic metre of air, $V_s$ is the velocity of the drop impact and $\beta$ is the impingement efficiency of the droplets, which is dependent on the aerofoil geometry and droplet diameter. The number of droplets per cubic metre is found from geometry and is presented by Springer as:

$$q = 530.5 \frac{I}{V_t d^3},$$ (11)

where $V_t$ is the terminal velocity of the droplets.

The rate of damage, $\dot{D}$, from a given precipitation intensity is found from:

$$\dot{D} = \frac{\dot{N}}{N_{ic}},$$ (12)

The analysis presented here has shown that the Best DSD currently used in the Springer model overestimates the size of impinging offshore droplets.

The exact number of impacts to failure is dependent on the protection system and substrate used. For a commercial erosion resistant polyurethane coating system, the offshore DSD has been applied to the above equations and the relative effect on leading edge erosion prediction of the DSD in relation to the Best DSD is presented in Figure 12.

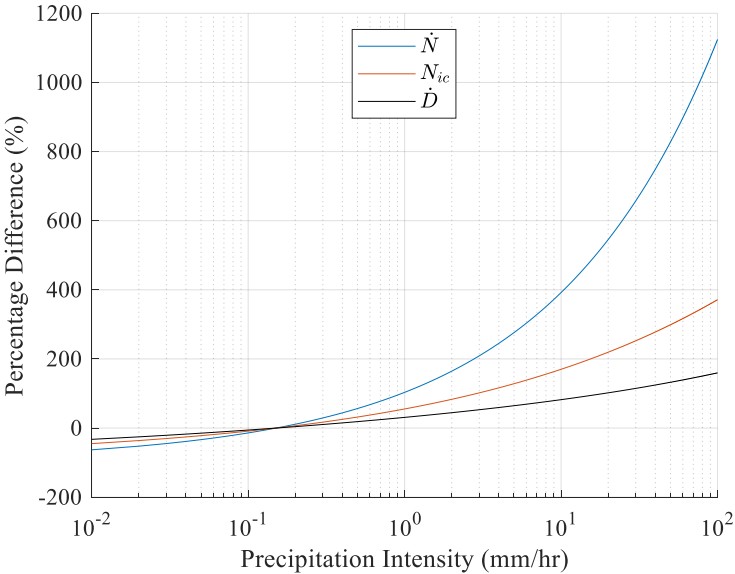

**Figure 12: Percentage change in leading edge erosion damage values from implementing the offshore DSD relative to implementing the Best DSD.**

The smaller median droplet diameter for precipitation intensities above 0.15 mm/hr requires a greater number of impacts to reach initiation. However, the equations show that there are a far greater number of droplet impacts per second, giving a higher damage rate for precipitation intensities, with the difference becoming substantial at the higher intensities. The impact of this is dependent on the site conditions and frequency of precipitation intensities. However, for the dataset presented here and the above material properties, the implementation of the offshore DSD causes the Springer model to predict a 23.7% reduction in lifetime in comparison to when the Best DSD is implemented. As a result, employing the Best DSD in leading

edge erosion prediction models underestimates the severity of the offshore environment in terms of leading edge erosion. Therefore, the lifetime of protection systems installed offshore is greatly overestimated, resulting in earlier than expected

maintenance and ultimately a higher cost of energy.

This dataset can be used to help to inform the lifetime of leading edge erosion protection systems installed offshore, helping to ensure maintenance is conducted early and further leading edge erosion can be combatted. The dataset can also be used to inform droplet impact models and rain erosion testing with the greater understanding of the environment facilitating the development of improved protection systems.

**7 Conclusions**

DSDs are important in predicting and modelling leading edge erosion. Currently, there is a lack of an offshore dataset and the industry uses onshore distributions in lifetime predictions. In this work, a disdrometer has been positioned three nautical miles offshore to collect and characterise the offshore precipitation environment and to provide an offshore DSD for lifetime prediction models.

Heavy and violent precipitation was rare in the measurement period, accounting for less than 3.5 hours of precipitation across the year. Therefore, erosion damage is not likely to be driven exclusively by heavy and violent precipitation. Rain was the most frequently occurring hydrometeor, whereas snow, ice and hail particles were scarce. A clear distinction was visible in the diameter-velocity plots for each hydrometeor, with snow particles occurring across a wider range of diameters and lower average velocities. The majority of raindrops observed had a diameter below 2 mm.

A general offshore DSD has been presented. The raw data was compared to the presented DSD and the most widely used DSD proposed by Best. A statistical $R^2$ analysis found that the offshore DSD aligned well with the data, whereas the Best DSD significantly overestimated the diameters of droplets. The implication of the offshore DSD was evaluated with the Springer model where it was found the inaccuracies in the Best DSD greatly underestimates the severity of the offshore environment in terms of leading edge erosion. As a result, the Best DSD is not a suitable distribution to use in lifetime

prediction models for protection systems positioned offshore and therefore predictions determined using it are unlikely to be accurate.

The results presented address the lack of an offshore dataset and provide a general offshore DSD that can be used to inform lifetime prediction models for the offshore environment. A disdrometer has been placed at ORE Catapult's Levenmouth offshore wind turbine to provide further information about the precipitation environment and validate the presented DSD.

The offshore dataset can be used to improve prediction and modelling techniques, helping to inform the design of new protection solutions and help combat leading edge erosion, whilst reducing lost energy production and unexpected turbine downtime.

Data availability. Please contact the corresponding author.

Author contributions. RH had the lead on paper writing, data analysis and derived conclusions. KD and PH were responsible for installing and setting up the disdrometers. KD and CW supervised the research.

Competing interests. The authors declare that they have no conflict of interest.

Acknowledgments. This work was supported by the Engineering and Physical Sciences Research Council through the EPSRC Centre for Doctoral Training in Composites Manufacture (Grant: EP/K50323X/1), project partners the Offshore Renewable Energy Catapult <https://ore.catapult.org.uk>, and the EPSRC Future Composites Manufacturing Hub (grant: EP/P006701/1). The authors would also like to thank the Wind Blade Research Hub for their support in delivery of this manuscript. All data necessary to reproduce the results and support the conclusions are included within this paper.

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
