# Peer review of "Characterisation of the Offshore Precipitation Environment to Help Combat Leading Edge Erosion of Wind Turbine Blades"

_Wind Energy Science, 2020_

## Referee Comment (RC1) · Anonymous Referee #1 · 15 May 2020

The paper contributed by Herring et al. from the title appear relevant for the journal. The offshore environment precipitation climate is a relevant research topic in perspective of leading edge erosion of turbine blades. The paper focuses on the meteorological side of this topic and in particular, compares the new drop size distribution data set versus the Best model published in 1950.

There are serious shortcomings in the paper. These can be categorized into

1) missing background information and discussion on precipitation meteorology.

2) insufficient presentation of the data processing, quality control and details on the instruments used.

[Figure]

3) statistical significance testing of the results.

4) relevance or implication of the new 'constants' and how these will influence assessment of precipitation in regard to leading edge erosion.

5) discussion of results is missing and the conclusion is a confusing mix of motivation and brief mention of some analysis results.

The title does not reflect the content of the article. There is no analysis of or description of how the observed rain data connects to combatting leading edge erosion.

These serious shortcomings are reason for rejection.

Below is given detailed review, in case the authors choose to improve the article and re-submit to a hydrological or meteorological journal.

Line 29. It says that Weather radars are widely used to predict the offshore precipitation environment. Please provide references to this and substantiate the entire paragraph on the background to your study. The text is short and unclear.

Line 34. It says that Kathiravelu et al. 2016 find Best to be outdated. This is not clear. The referenced work is a review on drop-size distribution measurement techniques during time. Along this line, it is noted that the sensor used in the current study is not in the list of Kathiravelu et al. The reference Agnew 2013 is mentioned and referenced as raindrops below 0.8 mm are slightly underestimated. Is that the only study available using this sensor? It is relevant to provide insight to the type of data collected versus other relevant recent data sets. The methods from 1950 are obviously not in use any more so the details on this appear out of scope for the current investigation.

The critical perspective on the Best function need a review of existing literature on this subject. This is missing from the article.

The drop size distribution is observed offshore in the North Sea. It would be relevant to cite and discuss other offshore drop size distribution data sets, e.g. from research

ships and other offshore sites (small islands), as well as coastal and land observations of drop size distribution in the UK. This information would be relevant as background information and introduction.

Does a weather radar cover the offshore site already with rain information available? In case, yes it would be interesting to have a brief background on this and the methodology in use (assuming it is Best model). Please add references.

Line 62. It says that two disdrometers are mounted, one at 25 m and the other at 55 m. Which of the two data series is presented in the current work? In the quality control section, it would be interesting to understand if both instruments observe similar precipitation and if quality check was done comparing the two time series. Looking at the photographs it appears that the flow field is different at the two heights. In case wind speed data are available it would be relevant to see if there is systematic influence as function of wind speed and wind direction to drop size and fall velocity between the two instrument's observations. One instrument is positioned vertical and the other horizontal.

Why? Table 1. Does this table include both liquid and solid precipitation, raw data before quality control?

Quality control is mentioned in subsequent section, so this is confusing. Are only data presented after this section quality controlled?

Line 81. The quality control appears too limited. It is recommended to ensure detailed quality control before subsequent analysis (Hasager et al. 2020 Renewable Energy). In particular, the hydrometeor-type frequency you present lines 114-137 was that included in the quality control? Did you use information on temperature to quality control hydrometeor type?

Figures 2 and 3. Do they include solid precipitation?

Lines 101-103. The seasonal breakdown of precipitation could be discussed later on

the discussion section, e.g. stratiform and convective events, and the influence to drop size distribution and rain intensity.

Line 108-110. The data is for one specific year. Annual variations are to be expected. So minutes and hours "a year" is misleading. Was this a wet year or a dry year?

Line 115. please refer to work on hail and leading edge erosion, e.g. Letson et al. 2020 WES, MacDonald et al. 2016 Wind Energy.

Line 115. In Bech et al., 2018 the rain intensity data were deduced from Jones and Sims, 1978, Maritime-temperate rain intensity frequency data. According to this data, a rain intensity of 10 mm/hr is exceeded approximately 0.06% of the time. This was rounded up to 0.1% in the presented model. 20 mm/hr was exceeded 0.02% of the time, and 50 mm/hr was exceeded 0.002% of the time. These numbers seem to be same size of order as what is reported in the present paper in review. Still the model and analysis presented in Bech et al. 2018 showed, that these few hours of heavy rain could cause the majority of damage observed on WT blades. However, the kinetic energy impact damage model probably over-estimates the effect of the droplet size, and thus the effect of rain intensity.

Line 118. A high amount of 'error' and 'unknown' occur (17.93%). It would be interesting to know if both instruments suffer 'equal' amounts of these and you could do 'gap-filling' from one instrument to the other, or find out what might be happening. In Table 1 it looks as July has most missing data. The total says 82.89% data that gives 17.11% missing data. Please clarify the numbers.

Figure 5. It is difficult to see the difference between snow grains and snowflakes with the colours chosen. Maybe use variation (not all open circles). It would be relevant to discuss the findings. How do you find your results are? As expected and reported elsewhere in literature? Give references in the discussion of results.

Figure 6. A suggestion is to use four different colours/symbols and put data into one

graphics. This would enable more clear reading of the data set and make it possible to see the lines be different. Furthermore, statistical test on significance of your results are necessary to draw conclusions. This also goes for Figs.7 and 8.

Table 4. You could include a row with the constants from Best 1950, i.e. merge table 3 and table 4.

Line 207. It says "Not appropriate to validate offshore weather radar data against Best DSD".

It is unclear what you mean. Please clarify. It would be relevant to include reference to the work you have in mind stating this sentence, and explain the implication.

Line 215- 216. A section with discussion of the results versus state of the art research on the topic drop size distribution is necessary to include in the paper. It is also advisable to include perspectives on the drop size distribution and the impingement to turbine blades.

It is briefly mentioned (lines129-130) but not elaborated further. This would be a relevant perspective to discuss in the discussion section. The title of the paper says that you are studying leading edge erosion but you do not bring your data set into any perspective on this. So you will have to include that otherwise the title of the paper is misleading, and will need modification to reflect the content of the paper properly.

Line 227. It says "The offshore DSD aligned well with the data". It is unclear what is meant. Please clarify.

The conclusion is a mixture of background, very brief sentences on the actual work, and very long part on future perspective. It would be beneficial to ensure the conclusion major part is related to the learnings from the current research.

List of references It is too short with lack of relevant meteorological literature.

---

## Referee Comment (RC2) · Anonymous Referee #2 · 29 Jun 2020

This paper is significant and supplements a research topic treated by different authors to model and predict leading edge erosion of wind turbine blades. It is dedicated on a critical industrial and scientific challenge for wind industry nowadays. The paper is focused on the offshore precipitation environment characterization with the motivation of offering appropriate offshore droplet size distribution (DSD) as erosion lifetime predictions input data. The work also ponders results with particular approaches found on the literature.

The title and the abstract point out well the intention of the manuscript but the work lacks valid analysis or discussion in terms of its application on leading edge erosion

lifetime modelling. I suggest specifying on the paper title its focus on the accuracy for the quantification of droplet size distribution in offshore conditions, which is an important improvement of great value for the scientific and industrial community. The paper does not propose any connection of the severity of erosion through the expected lifetime, even when its apparently focused on such influences. I recommend positively to complete the work on this analysis for possible scientific or industrial use.

The document is well structured (many other possibilities could be also possible) and states clearly the scope and methodology. Introduction and references discussion improvement is necessary in order to set the limits of the specific offshore application. Literature reviews of well-known Best model is used to pointing out the weakness or strengths of other authors proposals, but one can achieve valuable recommendations and likely directions for the essential improvements of the comparing results. The authors refer with assessed particular experimental data different results comparing with Best model and their proposed offshore DSD model. In order to categorize the results as a new model definition to be used in lifetime prediction methodologies, a unique location case and a unique year-season is used. I recommend completing the work on the statistical validation of testing results with other raw data sources comparing the presented model with the original one and the reasons for such extensiveness and validation.

I recommend this manuscript for publication after revision required. There have been outlined some recommendations to the authors to be considered.

---

## Author Comment (AC1) · 6 Aug 2020

Please find the response to the reviewer comments and a tracked changes version of the manuscript in the attached document.

Please also note the supplement to this comment:
https://wes.copernicus.org/preprints/wes-2020-11/wes-2020-11-AC1-supplement.pdf

---

## Author Response (AR1)

**The paper contributed by Herring et al. from the title appear relevant for the journal. The offshore environment precipitation climate is a relevant research topic in perspective of leading edge erosion of turbine blades. The paper focuses on the meteorological side of this topic and in particular, compares the new drop size distribution data set versus the Best model published in 1950.**

Thank you for the positive feedback.

**There are serious shortcomings in the paper. These can be categorized into**

**1) missing background information and discussion on precipitation meteorology.**

This paper evaluates the Best DSD, which is used in the current prominent leading edge prediction models and evaluates whether it is suitable to be used in the offshore environment. Further background information on lifetime prediction modelling and the relevance of the research to industry has been included in *Introduction*. This serves to demonstrate the importance and value of the subsequent analysis. The following has been included:

"*The aim of the industry is to develop a methodology that can predict the lifetime of a protection system on a wind turbine from rain erosion tests. The DNV-GL project COBRA aims to address this, and Eisenberg proposes using the Springer model. Due to the lack of an offshore dataset, the project uses the onshore Best distribution published in 1950 (Best, 1950).*"

The precipitation methodology discussion throughout has been expanded to include a number of studies (Montero Martinez, 2016, Johanssen, 2020, Gossard, 1992, Brandes, 2002). We believe that this apparent shortcoming is perhaps due to the aim of the manuscript not being clearly explained. This has been addressed in the above changes to the *Introduction*.

**2) insufficient presentation of the data processing, quality control and details on the instruments used**

Thank you for pointing out this shortcoming and your detailed comments below. We have expanded the Section *4.1 Quality Control* to present the data processing and quality control in more detail, as well as enlarge the reference base.

Further information has been provided on the instruments used to provide the reader with greater knowledge about their setup and their method of recording precipitation. The following has been added in Section *3 Offshore Measurement Technique*:

"*Each disdrometer consists of two photodiode sensing heads, one near-IR diode laser head and one CS215 temperature and humidity sensor. The sensor heads are positioned 20° off-axis to the system unit axis, introducing a time-lag between the two sensors that enables the fall velocity and size of particles to be calculated.*"

**3) statistical significance testing of the results.**

Thank you for this comment. Ultimately, the final statistical check is how well the proposed DSD fits the offshore environment, which has been presented in Figure 10. A statistical $R^2$ check is completed in Figure 11 and the proposed DSD displays very good correlation to the offshore data illustrated by $R^2$ values greater than 0.95 across all precipitation intensities. To assess the robustness of every step could be a paper in its own right and it is only necessary to assess the endpoint. It is noted in the manuscript that the slight reduction in $R^2$ at higher intensities can be attributed to the reduced amount of heavy and violent precipitation recorded.

It is recognised in Section *5.4 Limitations* that the DSD presented is only applied to the one set of offshore data and to validate it, the distribution needs to be applied to another set of offshore data. As far as the authors are aware this is the only offshore distribution that is presented, and it is hoped from publication that others will be able to evaluate the DSD against their data. Offshore data is being collected at ORE Catapult's Levenmouth offshore demonstration turbine to validate the DSD.

**4) relevance or implication of the new 'constants' and how these will influence assessment of precipitation in regard to leading edge erosion.**

Thank you for pointing out this shortcoming and for the relevant detailed comments below. We have included the section in the manuscript: *6 Impact of DSD on Leading Edge Erosion Lifetime Prediction*. This section assesses the impact of the presented DSD against the Best DSD in terms of lifetime prediction. It is found that the Best DSD underestimates the severity of the offshore environment and the inclusion of the offshore DSD reduces the lifetime of a protection system by 23.7%.

**5) discussion of results is missing and the conclusion is a confusing mix of motivation and brief mention of some analysis results.**

Thank you for this point and your follow up points on the conclusion. A discussion of the results in terms of leading edge erosion has now been expanded on in Section *6 Impact of DSD on Leading Edge Erosion Lifetime Prediction*. The below changes have been made to the conclusion to ensure that the major part of the conclusion is focused on the key findings of the paper. We believe that this is reflected in lines 339 to 343, where the key finding of the paper – that the Best DSD is unsuitable for use in offshore lifetime predictions – is clearly stated.

*"The implication of the offshore DSD was evaluated with the Springer model where it was found the inaccuracies in the Best DSD greatly underestimates the severity of the offshore environment in terms of leading edge erosion. As a result, the Best DSD is not a suitable distribution to use in lifetime prediction models for protection systems positioned offshore and therefore predictions determined using it are unlikely to be accurate."*

**The title does not reflect the content of the article. There is no analysis of or description of how the observed rain data connects to combatting leading edge erosion.**

Please see our response to comment "4) relevance or implication…". The manuscript has been updated to include greater analysis on the relevance between the observed dataset and how it helps to combat leading edge erosion.

A paragraph explaining the benefit of this dataset and its role in combatting leading edge erosion is presented in Section *6*:

*"This dataset can be used to help to inform the lifetime of leading edge erosion protection systems installed offshore, helping to ensure maintenance is conducted early and further leading edge erosion*

*can be combatted. The dataset can also be used to inform droplet impact models and rain erosion testing with the greater understanding of the environment facilitating the development of improved protection systems."*

**These serious shortcomings are reason for rejection. Below is given detailed review, in case the authors choose to improve the article and re-submit to a hydrological or meteorological journal.**

Thank you for your review and opinion on this. However, we must disagree and believe that the manuscript is suitable for publication. We have listed below the criteria against which Wind Energy Science assess manuscripts and outlined the reasons for why this manuscript meet them.

1. Scientific significance

As is pointed out by both reviewers, leading edge erosion is a significant challenge for the offshore wind industry. Currently, there is a lack of knowledge of the offshore environment and the amount, type and intensity of precipitation has not been quantified (Hasager, 2020). The manuscript is novel and presents the first offshore precipitation dataset, addressing the shortcomings and meeting a clear research need. The aim of the industry is to predict the lifetime of a protection system from rain erosion tests. This is being pursued in the DNV-GL joint industry project COBRA. In this manuscript, further information has been included to outline how the offshore dataset can be used to inform the lifetime prediction models. The results presented offer substantial new data that can be used by other researchers to further develop models and validate their results, greatly reducing the uncertainty in the offshore precipitation environment.

2. Scientific quality

We acknowledge that the quality control section was too brief, and this has been expanded on to provide more detailed information about the scientific approach and applied methods. The approach taken to develop a general droplet size distribution (DSD) equation is clearly outlined and every step illustrated with the appropriate equations and figures. A statistical check against the offshore environment has been completed to evaluate the accuracy of the DSD. Therefore, there is sufficient information provided so that other researchers could repeat this work.

3. Presentation Quality

The scientific results are presented clearly and concisely, with appropriate figures included to outline the key results and methodology. Two slight suggestions are made to improve the readability of the figures and these have been updated in the manuscript. As is pointed out by Anonymous Reviewer 2, the manuscript is well structured with a clear flow from i) Introduction and problem, ii) methodology, iii) Results, iv) Discussion in terms of its application to leading edge erosion and v) Conclusions. No comments were made on the quality of written English by either reviewer.

As a result, the authors believe that this manuscript meets the review criteria and that the manuscript is suitable for publication.

We have addressed your further comments below.

**Line 29. It says that Weather radars are widely used to predict the offshore precipitation environment. Please provide references to this and substantiate the entire paragraph on the background to your study. The text is short and unclear.**

We are unable to publish any results in respect to the weather radars. The authors understand your comments and, as we unable to provide more information on this, have removed the reference to validating weather radars. The manuscript provides information on how the offshore dataset can be used to improve lifetime prediction methods and that is the focus on the manuscript. This is clearly stated in lines 30 to 33 and has been expanded on in Section 6 (see our response to comment "4) relevance or…"), bringing the paper into line with its title and abstract.

**Line 34. It says that Kathiravelu et al. 2016 find Best to be outdated. This is not clear. The referenced work is a review on drop-size distribution measurement techniques during time. Along this line, it is noted that the sensor used in the current study is not in the list of Kathiravelu et al. The reference Agnew 2013 is mentioned and referenced as raindrops below 0.8 mm are slightly underestimated. Is that the only study available using this sensor? It is relevant to provide insight to the type of data collected versus other relevant recent data sets. The methods from 1950 are obviously not in use any more so the details on this appear out of scope for the current investigation.**

The line points out that the manual measurement techniques used by Best – namely the stain method and the flour pellet method – are outdated and inaccurate, not that the Best DSD is outdated. Kathiravelu states on page 2 of the respective paper that "These very early, functional techniques were found to provide inaccurate results". The second reference to Kathiravelu in the manuscript is in respect to optical disdrometers, as is stated at the start of the sentence on line 74. Kathiravelu reports on page 8 of the same paper that "Optical technologies [] are non-intrusive rain drop techniques. These methods do not influence drop behaviour during measurement and have successfully resolved drop break up".

Thank you for pointing out the shortcoming in the review of other studies using the disdrometer. This has now been expanded on between lines 79 and 91 in Section *3 Offshore Measurement Technique*.

Whilst the methods from 1950 are not in use, the DSD resulting from them are. In terms of leading edge erosion, the leading lifetime prediction models (Eisenberg 2018, Springer 1979) implement the Best DSD. This point has been made clearer in lines 30 to 36. Therefore, it is appropriate to include a summary of the methods used by Best to provide background to the DSD which is utilised.

**The critical perspective on the Best function need a review of existing literature on this subject. This is missing from the article.**

The relevance of the Best DSD has been explained more clearly in the *Introduction* now. Lifetime prediction models currently implement the Best DSD to determine the damage caused by leading edge erosion. Critical studies of the Best DSD are all focused onshore. This study aims to evaluate its appropriateness offshore and therefore it is difficult to draw comparisons between any results from other studies. The fact that the Best DSD is implemented onshore and there may be more appropriate DSDs for onshore is outside the scope of this study.

**The drop size distribution is observed offshore in the North Sea. It would be relevant to cite and discuss other offshore drop size distribution data sets, e.g. from research ships and other offshore sites (small islands), as well as coastal and land observations of drop size distribution in the UK. This information would be relevant as background information and introduction.**

To the authors knowledge this is the first study into the offshore precipitation environment that has been published and therefore the results are completely novel. Hasager states that "Quantitative

knowledge on rain events at offshore wind farm sites is lacking in Denmark and elsewhere." (Hasager, 2020). Whilst coastal studies exist, we do not believe that they are relevant to this paper and introducing them would have little benefit as no effective comparison could be completed. The aim of the paper is to evaluate the Best DSD, which is currently used in lifetime prediction methodologies, against the offshore environment. At most, the introduction of another onshore dataset (that is not currently used by the wind industry) could show that it predicts the offshore environment better than Best. At worst, it would serve to confuse and dilute the message of the paper.

**Does a weather radar cover the offshore site already with rain information available? In case, yes it would be interesting to have a brief background on this and the methodology in use (assuming it is Best model). Please add references.**

Please see our response to comment "Line 29. It says…".

**Line 62. It says that two disdrometers are mounted, one at 25 m and the other at 55 m. Which of the two data series is presented in the current work? In the quality control section, it would be interesting to understand if both instruments observe similar precipitation and if quality check was done comparing the two time series. Looking at the photographs it appears that the flow field is different at the two heights. In case wind speed data are available it would be relevant to see if there is systematic influence as function of wind speed and wind direction to drop size and fall velocity between the two instrument's observations. One instrument is positioned vertical and the other horizontal. Why?**

In respect to quality control, please see our response to comment "2) insufficient presentation…".

Whilst the investigation into the wind speed would be interesting, this has not been published due to proprietary reasons and is outside the scope of this manuscript. A follow on paper considering these aspects is in discussion.

On quick glance, it does appear that the disdrometers are orientated differently, however, both instruments have been installed in the same orientation. Regretfully, due to the height and the fact that the disdrometers are offshore, it is challenging to obtain a better camera angle to display this.

**Table 1. Does this table include both liquid and solid precipitation, raw data before quality control? Quality control is mentioned in subsequent section, so this is confusing. Are only data presented after this section quality controlled?**

Thank you for pointing out this shortcoming in the manuscript. We acknowledge that this was confusing and have moved Table 1 to line 140 and have updated line 132 to include "*quality controlled*" to note that the data is after quality control.

**Line 81. The quality control appears too limited. It is recommended to ensure detailed quality control before subsequent analysis (Hasager et al. 2020 Renewable Energy). In particular, the hydrometeor-type frequency you present lines 114-137 was that included in the quality control? Did you use information on temperature to quality control hydrometeor type?**

Please see our response to comment "2) insufficient presentation…"

**Figures 2 and 3. Do they include solid precipitation?**

Both figures include solid precipitation. All precipitation is included up to the end of Section 4. At the start of Section 5 solid precipitation is removed. This is made clear by line 208 which states "*Only data where rain particles were the modal hydrometeor were examined.*"

**Lines 101-103. The seasonal breakdown of precipitation could be discussed later on the discussion section, e.g. stratiform and convective events, and the influence to drop size distribution and rain intensity**

Following your detailed comments, the paper focus has more clearly shifted to the implications of the dataset to lifetime prediction. Whilst it could be interesting to go into further detail on seasonal breakdowns and stratiform and convective events, the authors believe that is outside of the scope of this study.

**Line 108-110. The data is for one specific year. Annual variations are to be expected. So minutes and hours "a year" is misleading. Was this a wet year or a dry year?**

The average rainfall for the region is 650 mm a year. Therefore, this was a relatively dry year and the following line has been added at line 137 to present this information to the reader:

"*Including the missing data provides an annual accumulation of 500 mm, which is lower than the 650 mm average annual precipitation reported in Northumberland (WeatherSpark, 2020), indicating that the measurement year was a relatively dry year for the area.*"

With the inclusion of this line now serving as a reference, we believe that it is of interest to include the number of minutes and hours a year to demonstrate the point that heavy and violent rain accounts for a low percentage of the rainfall and that a turbine would likely experience very little before erosion occurs.

**Line 115. please refer to work on hail and leading edge erosion, e.g. Letson et al. 2020 WES, MacDonald et al. 2016 Wind Energy.**

Letson and MacDonald present very interesting work on the hail environment. However, in this dataset, ice pellets, hail and graupel consisted of only 0.94% of the hydrometeors, with very few events where ice pellets were the modal hydrometeor and none where hail was. Therefore, using the results from this study, it is challenging to state with confidence the impact of hail and this has not been considered. This is something that we will continually revisit as the disdrometers collect more data.

**Line 115. In Bech et al., 2018 the rain intensity data were deduced from Jones and Sims, 1978, Maritime-temperate rain intensity frequency data. According to this data, a rain intensity of 10 mm/hr is exceeded approximately 0.06% of the time. This was rounded up to 0.1% in the presented model. 20 mm/hr was exceeded 0.02% of the time, and 50 mm/hr was exceeded 0.002% of the time. These numbers seem to be same size of order as what is reported in the present paper in review. Still the model and analysis presented in Bech et al. 2018 showed, that these few hours of heavy rain could cause the majority of damage observed on WT blades. However, the kinetic energy impact damage model probably over-estimates the effect of the droplet size, and thus the effect of rain intensity.**

How the results are interpreted depends on the damage model used. The dominant industry model is the Springer model (Springer 1979, Eisenberg 2018) and not a kinetic energy model. This has now been made clearer in the Introduction. When the Springer model is applied to these results, most

damage would be caused by the low and medium intensities. "*When considering the Springer model,*" has been added to the start of this line to clarify the damage model assumed.

Longitudinally overtime the question of whether erosion is driven solely by high intensity rain can be related to erosion data to unequivocally answer the question – that is if industry data can be at all presented given its highly confidential nature. Currently, the industry only uses the Springer model and therefore this is the appropriate damage model to assume. The point has been made to highlight that there is uncertainty around this by the inclusion of the word "*suggests*".

**Line 118. A high amount of 'error' and 'unknown' occur (17.93%). It would be interesting to know if both instruments suffer 'equal' amounts of these and you could do 'gap-filling' from one instrument to the other, or find out what might be happening. In Table 1 it looks as July has most missing data. The total says 82.89% data that gives 17.11% missing data. Please clarify the numbers.**

With respect to the gap filling and clarifying the numbers, please see our response to comment "2) insufficient presentation…". To further clarify the numbers the sentence has been updated in line 179 to "*'Errors' and 'unknown' particles accounted for 17.93% of the hydrometeors recorded. These may be caused by insects, particles between states or equipment failures and have been ignored in the subsequent analysis, with any records where they were the modal hydrometeor removed.*"

**Figure 5. It is difficult to see the difference between snow grains and snowflakes with the colours chosen. Maybe use variation (not all open circles). It would be relevant to discuss the findings. How do you find your results are? As expected and reported elsewhere in literature? Give references in the discussion of results.**

Thank you for pointing this out. Figure 5 has now been updated to improve the visibility of the snow grains. Instead of open circles for all the data, each hydrometeor has been assigned a different marker and the colour of the snow grains has been updated from cyan to black.

The data in the plot has been compared against the literature models for the terminal velocity of water particles and the results were in line with those predicted by Gossard (Gossard et al., 1992 and Brandes (Brandes et al. 2002). The following has been included to reflect this:

"The presented velocities for water particles are in line with those predicted in models by Gossard (Gossard et al., 1992) and Brandes (Brandes et al., 2002). The data presented in the above figure is used in the subsequent analysis to estimate the number of droplets that impact the blade per second and inform lifetime prediction models."

**Figure 6. A suggestion is to use four different colours/symbols and put data into one graphics. This would enable more clear reading of the data set and make it possible to see the lines be different. Furthermore, statistical test on significance of your results are necessary to draw conclusions. This also goes for Figs.7 and 8.**

Thank you for the suggestion. We have updated Figure 6 to include the data into one graphic with different colours and symbols distinguishing them.

In respect to the comment regarding the statistical significance of the results, please see our response to comment "3) statistical significance…"

**Table 4. You could include a row with the constants from Best 1950, i.e. merge table 3 and table 4.**

Table 3 presents the non-seasonal offshore DSD constants, whilst Table 4 presents the seasonal constants, with no mention of the Best constants. However, it is a good idea to merge Table 3 and Table 4 and this has been completed in the revised manuscript. The Best constants have a slightly different form (i.e. n instead of N and q) and therefore it sadly would not work to include them in the merged table as well.

**Line 207. It says "Not appropriate to validate offshore weather radar data against Best DSD". It is unclear what you mean. Please clarify. It would be relevant to include reference to the work you have in mind stating this sentence, and explain the implication.**

Please see our response to comment "Line 29. It says…".

**Line 215- 216. A section with discussion of the results versus state of the art research on the topic drop size distribution is necessary to include in the paper. It is also advisable to include perspectives on the drop size distribution and the impingement to turbine blades.**

Please see our response to comment "4) relevance or implication…"

**It is briefly mentioned (lines129-130) but not elaborated further. This would be a relevant perspective to discuss in the discussion section. The title of the paper says that you are studying leading edge erosion but you do not bring your data set into any perspective on this. So you will have to include that otherwise the title of the paper is misleading, and will need modification to reflect the content of the paper properly.**

Please see our response to comment "4) relevance or implication…"

**Line 227. It says "The offshore DSD aligned well with the data". It is unclear what is meant. Please clarify.**

This point is in relation to the statistical significance of the offshore DSD against the offshore dataset shown in Figure 11, where consistently high $R^2$ values were recorded. To clarify this, the line has been updated to "A statistical $R^2$ analysis found that the offshore DSD aligned well with the data, whereas the Best DSD significantly overestimated the diameters of droplets."

**The conclusion is a mixture of background, very brief sentences on the actual work, and very long part on future perspective. It would be beneficial to ensure the conclusion major part is related to the learnings from the current research.**

Please see our response to comment "5) discussion of…"

**List of references It is too short with lack of relevant meteorological literature.**

Thank you for pointing out this shortcoming. The inclusion of further literature on the use of the disdrometer in other studies and quality control has greatly increased the length of the literature review and the number of meteorological literature.
**This paper is significant and supplements a research topic treated by different authors to model and predict leading edge erosion of wind turbine blades. It is dedicated on a critical industrial and scientific challenge for wind industry nowadays. The paper is focused on the offshore precipitation environment characterization with the motivation of offering appropriate offshore droplet size distribution (DSD) as erosion lifetime predictions input data. The work also ponders results with particular approaches found on the literature.**

Thank you for the positive assessment.

**The title and the abstract point out well the intention of the manuscript but the work lacks valid analysis or discussion in terms of its application on leading edge erosion lifetime modelling. I suggest specifying on the paper title its focus on the accuracy for the quantification of droplet size distribution in offshore conditions, which is an important improvement of great value for the scientific and industrial community. The paper does not propose any connection of the severity of erosion through the expected lifetime, even when its apparently focused on such influences. I recommend positively to complete the work on this analysis for possible scientific or industrial use.**

Thank you for highlighting this. We have included the section in the manuscript: *6 Impact of DSD on Leading Edge Erosion Lifetime Prediction*. This section applies the presented DSD in lifetime modelling, assesses its impact against the Best DSD and discusses the implications. It is found that the Best DSD underestimates the severity of erosion in the offshore environment and the inclusion of the offshore DSD reduces the lifetime of a protection system by 23.7%. With the inclusion of this section, we now believe that the paper title aligns with the work presented.

**The document is well structured (many other possibilities could be also possible) and states clearly the scope and methodology. Introduction and references discussion improvement is necessary in order to set the limits of the specific offshore application. Literature reviews of well-known Best model is used to pointing out the weakness or strengths of other authors proposals, but one can achieve valuable recommendations and likely directions for the essential improvements of the comparing results. The authors refer with assessed particular experimental data different results comparing with Best model and their proposed offshore DSD model. In order to categorize the results as a new model definition to be used in lifetime prediction methodologies, a unique location case and a unique year-season is used. I recommend completing the work on the statistical validation of testing results with other raw data sources comparing the presented model with the original one and the reasons for such extensiveness and validation.**

Thank you for the feedback on the structuring.

Improvements to the introduction have been made to consolidate the aim and relevance of the study to lifetime prediction modelling. The following paragraph has been included:

"*The aim of the industry is to develop a methodology that can predict the lifetime of a protection system on a wind turbine from rain erosion tests. The DNV-GL project COBRA aims to address this, and Eisenberg proposes using the Springer model. Due to the lack of an offshore dataset, the project uses the onshore Best distribution published in 1950 (Best, 1950).*"

The number of references and their discussion has been expanded to include a number of relevant studies (Montero Martinez, 2016, Johanssen, 2020, Gossard, 1992, Brandes, 2002). In lines 79 to 90, greater detail has been provided on studies that have used the same disdrometer and their findings are reviewed. In line with comments from Anonymous Reviewer 1, the quality control section has been expanded, increasing the number of literature studies (see their comment starting "2) insufficient presentation").

In relation to comparison with other raw data sources, as far as the authors are aware this is the only offshore dataset that has been obtained and presented. Hasager states that "Quantitative knowledge on rain events at offshore wind farm sites is lacking in Denmark and elsewhere." (Hasager, 2020). There are available onshore and coastal datasets, however introducing them would have no little benefit as no effective comparison could be made. The aim of the paper is to evaluate the Best DSD, which is currently used in lifetime prediction methodologies, against the offshore environment. It is recognised in Section *5.4 Limitations* that the DSD presented is only applied to the one set of offshore data and to validate it, the distribution needs to be applied to another set of offshore data. Offshore data is being collected at ORE Catapult's Levenmouth offshore demonstration turbine to provide the validation data required. It is hoped from publication that others will be able to evaluate the DSD against their data.

**I recommend this manuscript for publication after revision required. There have been outlined some recommendations to the authors to be considered.**

Thank you for the feedback and taking the time to review the manuscript.

Please find the tracked changes version of the manuscript on the subsequent page. Please note that Mendeley has been used for reference formatting and changes to the references have not been marked up in the tracked changes document.

[revised manuscript text omitted]

---

## Author Response (AR2)

We are grateful to the editors for their time, feedback and decision on our manuscript. We have provided a response to the received comment. The comment is displayed in bold font, with our response below in non-bold font. A tracked changes version of the manuscript can be seen on the subsequent page.

**Associate Editor Decision: Publish subject to minor revisions (review by editor). Comments to the author**

**Fig. 10. The precipitation intensities in the graphics and the legend differ. Please clarify.**

Thank you for pointing this out. The figure caption has been updated to ensure consistency between the precipitation intensities shown.

[revised manuscript text omitted]